# Microfluidic Platforms for the Isolation and Detection of Exosomes: A Brief Review

**DOI:** 10.3390/mi13050730

**Published:** 2022-04-30

**Authors:** Duraichelvan Raju, Srinivas Bathini, Simona Badilescu, Anirban Ghosh, Muthukumaran Packirisamy

**Affiliations:** Micro-Nano-Bio Integration Center, Optical Bio-Microsystems Laboratory, Department of Mechanical Industrial and Aerospace Engineering, Concordia University, Montreal, QC H3G 1M8, Canada; duraichelvan.raju@concordia.ca (D.R.); bathini.srinivas3@gmail.com (S.B.); simonabadilescu0@gmail.com (S.B.); anirgho@gmail.com (A.G.)

**Keywords:** exosomes, isolation, detection, lab-on a chip, microfluidics devices

## Abstract

Extracellular vesicles (EVs) are a group of communication organelles enclosed by a phospholipid bilayer, secreted by all types of cells. The size of these vesicles ranges from 30 to 1000 nm, and they contain a myriad of compounds such as RNA, DNA, proteins, and lipids from their origin cells, offering a good source of biomarkers. Exosomes (30 to 100 nm) are a subset of EVs, and their importance in future medicine is beyond any doubt. However, the lack of efficient isolation and detection techniques hinders their practical applications as biomarkers. Versatile and cutting-edge platforms are required to detect and isolate exosomes selectively for further clinical analysis. This review paper focuses on lab-on-chip devices for capturing, detecting, and isolating extracellular vesicles. The first part of the paper discusses the main characteristics of different cell-derived vesicles, EV functions, and their clinical applications. In the second part, various microfluidic platforms suitable for the isolation and detection of exosomes are described, and their performance in terms of yield, sensitivity, and time of analysis is discussed.

## 1. Introduction

EVs are spherical particles enclosed by a phospholipid bilayer released from eukaryotic and prokaryotic cells. EVs are present in blood and body fluids such as urine, saliva, breast milk, and cultured media [1,2,3,4,5]. The typical diameter of exosomes, a subset of EVs, is between 30 and 100 nm, much smaller than red blood cells. It consists of DNA, RNA, mRNA, miRNA, proteins, nucleic acids, heat shock proteins (HSP70 and HSP90), RAB proteins that regulate docking and membrane fusion of EVs with recipient cells adhesion molecules (integrins and lactadherin), and the tetraspanins, such as CD9, CD63, and CD81. They facilitate intercellular communication and regulate crucial cell processes such as coagulation, inflammation, and cellular homeostasis [6,7,8,9,10,11,12,13]. There is a growing interest in the clinical applications of exosomes as biomarkers for disease diagnosis, therapy, prognosis, and diagnosis. Despite increasing scientific and clinical interest, no standard procedures are available to isolate, detect, and characterize exosomes because their size is below the reach of conventional detection methods.

Exosomes are released from the cell when multivesicular bodies fuse with the plasma membrane, or directly from the plasma membrane. Exosomes may be released in two ways, as shown in Figure 1a: firstly, the “classic pathway,” which involves the formation of intraluminal vesicles (ILVs) within the multivesicular endosomes (MVEs). They, in turn, fuse the membrane of MVE with either lysosome for cargo degradation or the plasma membrane, resulting in the release of ILVs called exosomes. The second way is the direct budding of the plasma membrane and is called the “direct pathway.” The extent to which such exosomes are released from other cells or in vivo (e.g., in biological fluids) is unknown. The other cell-derived vesicles are microvesicles and apoptotic vesicles, as shown in Figure 1b. 

Many excellent review papers on microfluidic detection have been published in the last decade [14,15,16,17]. However, they have not critically evaluated the performance of various devices in terms of the complexity of the devices and the results. The present review paper aims to discuss in depth the principle of the technique and its benefits for future clinical assays.

The immune properties of different exosomes suggest that they may be helpful as vaccines for infectious diseases [18]. Thus, exosomes may be potential sources for anticancer vaccines and may eliminate infections [19]. Therefore, exosomes are considered for clinical applications in treating ailments such as toxoplasmosis, diphtheria, tuberculosis, and typical severe acute respiratory syndrome and autoimmune diseases. As exosomes play a vital role in the drug delivery systems, researchers are trying to find applications for them in treating autoimmune/inflammatory diseases [20]. 

## 2. Exosomes as Reliable Biomarkers for Early Diagnosis of Cancer

As mentioned previously, extracellular vesicles are spherical particles enclosed by a phospholipid bilayer [21,22,23,24,25,26]. They are present in all biological fluids [27]. They play a crucial role in intercellular communication by transporting and delivering cargo between their cells, promoting disease progression. Exosomes and circulating tumour cells (CTC), and circulating tumour DNA (ctDNA) have attracted much attention in the past decades as biomarkers for the early diagnosis of cancer. The major challenge in the isolation of CTCs and ctDNA is that the isolation and analysis of CTCs are challenging as they are infrequent and heterogeneous [28]. At the same time, the ctDNA is not stable in the blood or the other body fluids and may be highly fragmented [29,30]. Because of these intrinsic limitations of CTCs and ctDNA, their definite isolation and precise detection remain challenging even though there have been many advancements in technology. Compared to CTCs and ctDNA, exosomes have a lot of advantages in terms of stability, quantity, and accessibility. Moreover, the concentration of exosomes increases tremendously (manyfold) between cancerous and non-cancerous cells as the tumour progresses when compared to the other biomarkers such as tumour antigen, CTC, and ctDNA. For example, in the case of a glioblastoma (GBM) patient plasma study, the concentration of exosomes was approximately 50 times higher than that of a healthy patient [31].

Further, from, Figure 2, it is noticeable that the expression level of exosomes is relatively higher than that of CTCs and tumour antigens in stage I, which is the early stage of cancer [32]. Exosomes are highly stable and capable of protecting nucleic acids and proteins closely related to cancer development. Many studies have shown that the cancer cell-derived exosomes contain specific nucleic acids and proteins that reflect the origin of cancer cells and the type of cancer. [33,34]. Therefore, exosomes can be potentially utilized for therapy, prognosis, and promising biomarkers for early cancer diagnosis. 

## 3. Isolation and Detection Techniques

The problem impeding the advancement of exosomes research is the standardization of sample collection protocols such as sample collection, sample processing, and sample analysis for translating exosomes to suitable clinical biomarkers. Exosomes are present in a wide range of body fluids. Low-speed centrifugation is enough for removing cells and large vesicles, but for pelleting exosomes, high-speed ultra-centrifugation is required [35]. Repeated ultracentrifugation steps can damage the exosomes and thus reduce their yield, potentially impacting their content’s proteomics and RNA analysis [36]. Therefore, both the International Society for Thrombosis and Haemostasis (ISTH) and the International Society for Extracellular Vesicles (ISEV) have described the guidelines and recommendations regarding the standardization of sample collection and handling protocols [37] as shown in Figure 3.

Despite the increasing scientific and clinical interest, no standard procedures are available to isolate, detect, and characterize exosomes because their size is below the reach of conventional detection methods. Given the growing evidence that exosomes may be a clinically relevant biomarker source, there is a great demand for their efficient and straightforward detection from bio-fluids. Most affinity-based methods rely on antibodies directed against exosome surface markers. Therefore, choosing the best protocol and customizing it based on the study seems necessary.

### 3.1. Exosomes Isolation Methods Based on Their Physical Properties

The traditional methods are presented in Figure 4 for exosome physical characterization and molecular analysis. The technique used to study the morphology of the exosomes is scanning electron microscopy (SEM) and atomic force microscopy (AFM). The size and concentration of exosomes can be determined by nanoparticle tracking analysis (NTA), dynamic light scattering (DLS), tuneable resistive pulse sensing (TRPS), or flow cytometry (FC). Nucleic acid quantification and analysis of exosome proteins can be done by methods such as bicinchoninic acid (BCA) assay, Western blotting (WB), enzyme-linked immunosorbent assay (ELISA), liquid chromatography-tandem mass spectrometry (LC-MS/MS), nucleic acid extraction, and polymerase chain reaction (PCR). 

Typically, the isolation methods can be classified into four types: density-based, size-based, surface component-based, and precipitation methods, as shown in Figure 5a. Density-based methods such as differential ultracentrifugation and density gradient centrifugation were developed during the early-stage research of exosomes. In differential ultracentrifugation, large cell debris and cells were initially removed at a low speed of around 20,000× *g*. After this, the proteins were removed by precipitating exosomes at higher speeds (higher than 100,000× *g*). At the same time, in density gradient centrifugation, a series of solutions with different densities are preloaded into a centrifugal tube before the addition of the sample. Exosomes can then be isolated via ultracentrifugation due to differences in the densities once the balance is achieved between centrifugal force and buoyancy. Expensive equipment and the more significant time requirements limit ultracentrifugation in the clinical context for isolating exosomes. In addition, these techniques include multiple steps and take around 5 to 6 h [38,39]. 

The isolation techniques based on size include membrane filtration, size exclusion chromatography (SEC), and other isolation methods that may be carried out in microfluidic chips. Filtration is a high-throughput and straightforward method in which the exosomes are isolated based on the size differences of the particles in the sample. Size exclusion chromatography (SEC) enables the separation of polymers or proteins based on their size or hydrodynamic volume. In this method, a porous matrix is usually packaged into a column as a stationary phase, as shown in Figure 5b. When a sample passes, the components smaller than the pore diameter can enter the porous material and take longer to pass through the column, while the larger ones cannot enter the pores.

Therefore, at different elution times, several components can be separated. Thus, this method extensively isolates exosomes [40,41,42]. However, there is a high chance that exosomes or contaminant aggregates will get trapped in the pores, possibly damaging them.

The polymer precipitation method for the isolation of exosomes ensures a high yield by simplifying the process and reducing the handling time. Companies have commercialized these polymer-based precipitation methods as ExoQuick, Total exosome isolation, and Exospin because of their simple protocols and fast isolation [43]. In this method, polymers are dissolved in the sample to reduce their solubility, because of which low-speed centrifugation can precipitate exosomes for isolation. Polyethylene glycol (PEG) is used as a precipitation agent to isolate exosomes [39,44]. Though this method is simple, fast, and high yield, exosomes lack purity as the protein aggregates and other contaminants may be co-precipitated. Besides this, it may change exosomes’ structure and surface characteristics [45]. These drawbacks can severely affect the further analysis of exosomes.

**Figure 4 micromachines-13-00730-f004:**
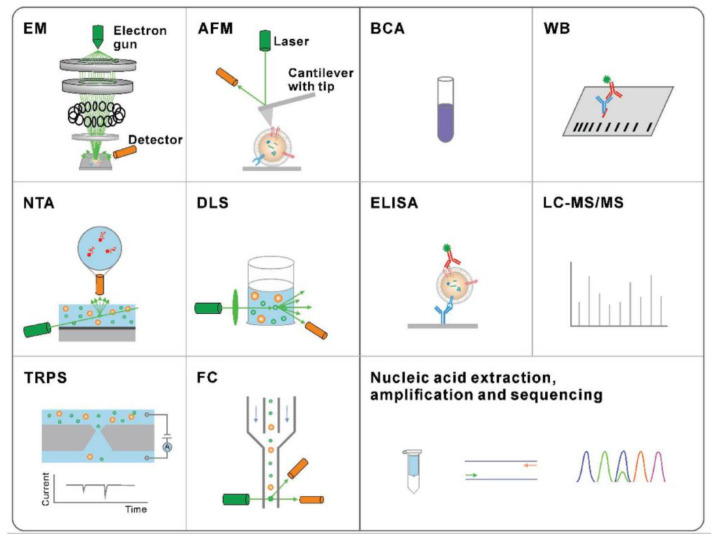
Traditional methods for the characterization of exosomes. Characterization techniques to measure physical properties (morphology, size, and zeta potential) (Reproduced with permission from Wang W. et al. [39]. Copyright 2018, John Wiley and Sons).

**Figure 5 micromachines-13-00730-f005:**
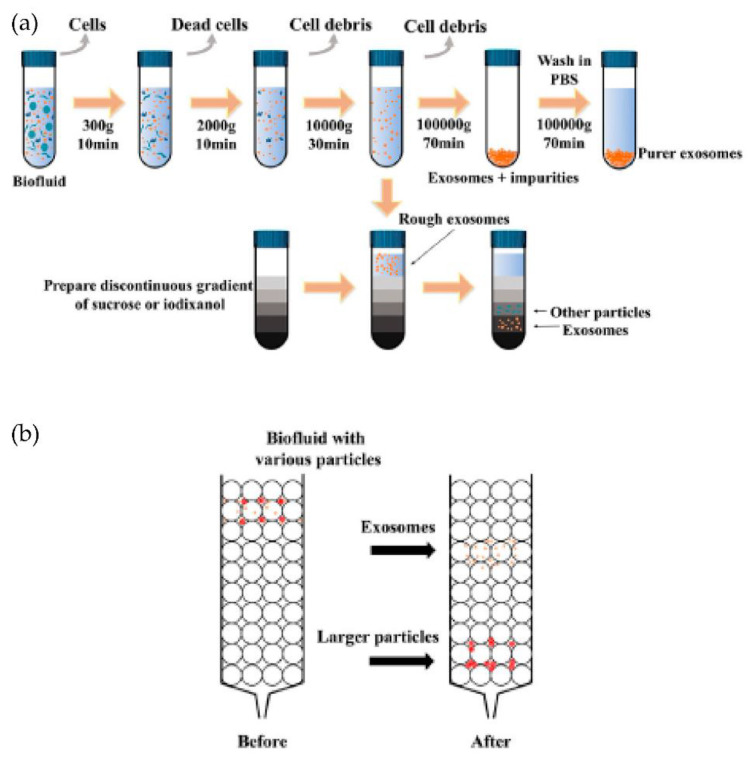
Schematics of the common exosomes isolation methods. (**a**) Ultracentrifugation and density gradient centrifugation. (**b**) Size-exclusion chromatography (SEC). (Modified from Chen J. et al. [42]. Copyright 2022, Front Bioeng Biotechnol).

The presence of many proteins and the lipid bilayer on the surface of exosomes make the immunoaffinity-based methods highly suitable for their isolation. The frequently used technique for specific capture and isolating exosomes is affinity-based isolation. These methods are based on binding antibodies or aptamers by a lipid probe, as shown in Figure 6. An antibody with affinity to exosome proteins is usually modified on a solid surface such as magnetic beads or a microfluidic channel to separate exosomes from a culture media or nonspecific vesicles and other contaminants [46]. In addition to beads or microfluidic chips, nanoparticles or nanomaterials may provide more suitable substrates to capture exosomes because of increased binding sites on their surface. Similar to antibodies, polypeptides can also be used as affinity agents to isolate exosomes [47]. Besides antibodies and peptides, the DNA aptamer can also separate exosomes, expressing a specific protein. The aptamer is a screened nucleic acid fragment with a particular sequence. Therefore, it can bind with a high affinity towards specific proteins. The usage of aptamers for the isolation of exosomes has benefits such as their low cost, high stability, and easy production [48]. Though aptamers have advantages over antibodies, their rare availability with a specific affinity to exosomes is their major drawback. To avoid the disadvantages of affinity-based isolation methods, an alternative approach to isolate the exosomes is to bind them with the lipid bilayer by designing a lipophilic isolation probe. Thus, capturing efficiency can be improved by ensuring the designed lipid probes have a high affinity toward exosomes’ lipid membranes. Wan et al. [49] reported using magnetic extraction to isolate exosomes in a short processing time of 15 min. The resulting exosome purity is higher than that obtained by ultracentrifugation, and the processing is faster. Further, the lipid probe isolation method can capture exosomes irrespective of their size and surface antigen, consequently escaping the loss produced by the surface marker.

### 3.2. Lab-on-a-Chip (LOC)/Microfluidics for Isolation and Detection of Exosomes

For an improved treatment and control of the progression of the disease, a rapid and early diagnosis is required. Common detection methods such as polymerase chain reaction (PCR) and enzyme-linked immunosorbent assay (ELISA) depend heavily on expensive and sophisticated equipment. Lab-on-a-chip (LOC) technology has emerged during the last two decades and has drawn significant interest for its biomedical applications. A microfluidic device or a lab-on a chip (LOC) may integrate conventional isolation methods by applying fluid dynamics principles. The advantages of LOC include high throughput, low sample and reagent consumption, short assay time, and multiplexed detection [50,51,52].

Further, the large surface-area-to-volume ratios accelerate heat and mass transport within the micro-channels and help rapid and controllable mixing, cooling, and manipulation of variables such as temperature, concentration gradient, and pressure. These merits are also crucial for various applications. LOC technology has shown the potential to improve biomarker detection by offering sensitive and wide-ranging measurements in a compact format.

Therefore researchers have recently started integrating different methods based on the physical properties of exosomes with the design of the microfluidic device. Thus, microfluidics-based isolation techniques have developed and evolved over the years [53,54,55,56,57,58,59,60,61,62,63,64,65,66,67,68,69,70,71]. The following sections will discuss some of the most interesting methods mostly based on the immunoaffinity approaches. Some are combined with nanoplasmonics detection and integrated with microfluidics to capture and detect the exosomes.

Immunoaffinity capture methods use immunoaffinitive interactions between antigens, exosome membrane proteins, and monoclonal antibodies. Exosome-specific proteins displayed on exosome membranes are principally tetraspanin proteins, for example, CD9, CD81, CD82, CD87, CD63, and heat-shock proteins produced in response to stress, as well as other proteins involved in cell adhesion and signalling. Immunomagnetic bead-based exosome isolation is one of the most common immunoaffinity-based capture methods that results in high-purity exosomes. In this case, the magnetic beads are coated with antibodies specific to the surface proteins of exosomes.

#### 3.2.1. Immunoaffinity Methods to Capture and Detect Exosomes

This approach has been frequently utilized to capture exosomes, and a number of suggested devices are briefly described below.

Chen et al. [31] designed an easy and rapid microfluidic immunoaffinity-based method to isolate microvesicles from small volumes of serum blood samples and conditioned medium from cells in culture. As shown in Figure 7, they developed two different microfluidic device designs with a straight flow channel for processing sample volumes of 400 μL and even smaller, with herringbone grooves on their ceiling. Their design achieved a 42–94% yield of exosomes with high specificity and shot isolation time. To improve the capture yield of the chip, the authors considered modifying parameters such as the dimensions of the microchannel, including structures inside the microchannel, increasing the transverse flow and the coverage of active antibodies. The main advantage of this design is that the device can sort the microvesicles directly from the serum in a single step.

A couple of years later, Wang et al. [72], designed and fabricated a microfluidic device based on porous silicon nanowires on a micropillar structure as shown in Figure 8. With the help of a prototype design, they could trap exosome-like lipid vesicles within 15 min while filtering out proteins and cell debris. Besides this, the trapped lipid vesicles can be recovered intact, with a yield of 67–60%, by dissolving the porous nanowires in a PBS buffer. Nonetheless, the recovery time is about one day, which is a critical concern.

Jorgensen et al. [73,74], using a non-contact printer, printed 24 microarray spots, as shown in Figure 9. The micro spots were printed with a cocktail of antibodies against the tetraspanins CD9, CD63, and CD81, selected to ensure that all exosomes were detected. With their approach, the authors could detect the exosomes with a sensitivity of 5000 particles/μL.

He et al. [75] developed microfluidic devices with serpentine channels and integrated specific immunoaffinity isolation of targeted proteins with immunomagnetic beads to isolate the exosomes directly from the human plasma, as shown in Figure 10. With their techniques, the isolation of a selective subpopulation of biomarkers from plasma samples (30 μL) was possible in 1.5 h, with markedly improved detection sensitivity and a yield of 42–97.3%. However, the method is specific only to CA125, EpCAM, and CD24 proteins. Due to the simplicity of their design, scaling up for the high-throughput screening of cancer and non-cancerous diseases is possible.

Kanwar et al. [76] designed and fabricated a simple, low-cost microfluidic-based device to isolate cirEVs enriched in exosomes directly from blood serum to simultaneously capture and quantify the exosomes within a single device. The schematic and the fluorescence detection scheme of the device are shown in Figure 11. The microfluidic device was fabricated using polydimethylsiloxane (PDMS) and then functionalized with antibodies against CD63, an antigen commonly overexpressed in exosomes. Subsequently, it was stained with a fluorescent carbocyanine dye (DiO) that labels the exosomes. The exosomes were quantified using a standard plate-reader. The authors achieved a yield of 15–18 μg of total proteins, and the scaling of their device is relatively easy.

Using a different approach, Lee et al. [77] incorporated an acoustic nano filter system in their design that separates EVs continuously, as shown in Figure 12. They used the separation of ultrasound standing waves to apply differential acoustic forces on MVs according to their size and density. By optimizing the design of the ultrasound transducers and the underlying electronics, they could achieve >90% separation yield and permitted in situ control of the size cut-off. To expand the functionality of their design, they suggested that several aspects of the system can be further developed, such as having different transducer designs to control the acoustic force and improve the sample throughput. Additionally, to realize a portable lab-on-chip for MV analyses integrating analytical components such as sensors and polymerase chain reaction into the same platform will assist clinical applications.

Zhao et al. [78] developed a microfluidic ExoSearch chip that is easy to operate to detect the three exosomal tumour markers (CA-125, EpCAM, CD24), as shown in Figure 13. With their design, the authors were able to detect the exosomes with a limit of 750 particles/μL, which is 1000 times more sensitive than conventional methods such as Western blotting.

A microfluidic device (nano-IMEX) with a single channel, as shown in Figure 14, was developed by Zhang et al. [79]. The nano-IMEX chip contains Y-shaped microposts functionalized with graphene oxide (GO), and the induced nanostructured polydopamine (PDA) film by the microfluidic layer-by-layer deposition, thus permitting simple covalent protein conjugation through the PDA chemistry. Through their technique, the authors have demonstrated that the nanostructured GO/PDA interface can significantly improve the efficiency of exosome immuno-capture, suppressing the nonspecific exosome adsorption. Based on this nano-interface, they could achieve a detection limit of 50 μL—substantially better than the existing methods.

A microfluidic device for the immunocapture of circulating exosomes from both cell culture medium and patient plasma from a low sample volume was developed Fang et al. [80] as shown in Figure 15. For capturing the exosomes, CD63 antibodies conjugated with magnetic nanoparticles (Mag-CD63) were used. Their study found that the amount of the exosomal tumour marker EpCAM was much higher in the plasma of breast cancer patients than in healthy controls. The authors concluded that the microfluidics device might become a valuable tool for breast cancer diagnosis due to advantages such as its better purity, intact yield, simplicity of operation, less time, and low cost.

More recently, our group [81] has developed a magnetic particle-based liquid biopsy chip to isolate EVs by using a synthetic peptide, Vn96, which specifically binds to heat shock proteins (HSP). In this design, a 3D mixer integrated within the chip, as shown in Figure 16, improves capture efficiency and a sedimentation unit that allows the EV-captured magnetic particles to settle based on gravity. The significant advantages of the label-free technique implemented using streptavidin-coated magnetic particles include faster isolation of EVs from CCM (around 20 min) and easy removal of magnetic particles using a magnet after elution. The maximum isolation efficiency obtained was about 90% with 8.0–9.9 μm streptavidin-coated magnetic particles when 0.2 mL CCM was used.

In the past few years, various research groups have used a nanoplasmonic approach integrated with microfluidics to enhance the capture and detection of EVs. Some of the nanoplasmonic methods are discussed briefly in the following section.

#### 3.2.2. Immunoaffinity Methods with Nanoplasmonic Detection of Exosomes

Im et al. [82] developed for the first time a nanoplasmonic exosome (nPLEX) assay based on the surface plasmon resonance principle. As shown in Figure 17, periodic nanohole arrays are the exosomes detection sites. They detected the exosomes derived from ovarian cancer cells, expressing CD24 and EpCAM. Therefore the nanohole arrays were functionalized with antibodies against CD63. The detection sensitivity of the nPLEX assay was determined to be 1000 particles/μL, 10^4^ times higher than the Western blotting and 10^2^ times higher than the ELISA method.

A multiplexed microfluidic device based on tuneable alternating current electro hydrodynamics (acEHD, nanoshearing) was developed by Vaidyanathan et al. [83], as shown in Figure 18. Using their device, they detected exosomes derived from cells expressing the human epidermal growth factor receptor 2 (HER2), prostate-specific antigen (PSA), and CD9. Through their method, the detection sensitivity of 2760 exosomes/μL was achieved, which means a threefold enhancement, compared to hydrodynamic flow-based assays (8300 exosomes/μL).

Zhu et al. [84] presented a real-time, label-free method to detect and characterize tumour-derived exosomes in CCS without enrichment or purification, as shown in Figure 19. They used surface plasmon resonance imaging (SPRi) with antibody microarrays specific to exosome transmembrane proteins such as CD9, CD63, and CD81. With this approach, they could achieve a detection sensitivity of 4.87 × 10^4^ particles/μL.

Our group [85,86] developed a simple microfluidic device to detect MCF-7 exosomes using an immune-affinity approach, Vn96 polypeptide, and LSPR detection. The schematic of the device and the sensing protocol are shown in Figure 20. Their results indicate that the label-free technique, based on the sensitivity of the Au-LSPR band to the surrounding environment, is a promising approach. The Au nano-island platform can capture a high number of extracellular vesicles present in the MCF7, providing a detection range covering early stages to advanced stages of cancer. However, the authors did not mention the sensitivity of detection of their method.

Raghu et al. [87] developed a nanosensing array, combining the nano and microfabrication techniques for single exosome detection. As shown in Figure 21, plasmonic nanopillars were fabricated to accommodate at most one exosome. Therefore, to target the tetraspanins CD63 membrane-bound proteins in exosomes secreted by MCF7 breast adenocarcinoma cells, the plasmonic nanopillars were functionalized with anti-CD63 antibodies. Using this approach, the authors could achieve sensitivity improvements three orders of magnitude over previously reported real-time, multiplexed platforms.

A plasmonic platform based on gold nano-ellipsoids with an integrated microfluidic device was developed by Xiaoqing Lv et al. [88]. The nano-ellipsoids were fabricated using an anodic aluminium template as a shadow mask. The nano-ellipsoids’ surface was functionalized with anti-CD63, specific to exosome transmembrane proteins. Figure 22 shows the schematics of the sensing protocol. The authors claim that the uniform adsorption of exosomes on the functionalized nano-ellipsoids enhanced the detection sensitivity of their design.

Based on the markers used for detection, the yield sensitivity, the yield capability, advantages, disadvantages, and the year the work was published, some of the most interesting immunoaffinity and nanoplasmonic approaches integrated with microfluidics-based techniques are summarized and tabulated in Table 1 and Table 2. The tables show that the microfluidic/LOC technology developments have enhanced exosome capture and detection in terms of their efficiency, compared to previous complicated and time-consuming methods.

It is not easy to compare the methods discussed in Section 3.2 and summarized in Table 1 and Table 2 because of their complexity and diversity. Several of them can detect exosomes in a reasonable timeframe, with high sensitivity and a good yield, but their design is generally not straightforward. Many published methods involve complex structures such as microposts, micro and nanopillars, or nano-ellipsoids, requiring complex microfabrication methods. Some of the techniques use printed arrays that can be fabricated easily. The results in terms of sensibility are very promising. Good results, especially regarding exosomes isolation, have also been obtained using functionalized magnetic particles.

## 4. Outlook

Microfluidic isolation and detection methods represent clear progress compared to conventional strategies. Nevertheless, first-generation devices like those described in this review paper are not yet ready to be translated into clinical analysis. The main reason for this is the lack of standardization and validation of the microfluidic methods and the relatively low processing capacity. Additionally, because of the complexity of all the biological samples, the overlap of size between exosomes and other EVs, and the heterogeneity of exosomes, the purity of the isolated exosomes is lower. For this reason and other considerations, conventional methods like ultracentrifugation still account for 56% of all exosome isolation techniques employed in exosome research. However, despite the large sample capacity and high yield, the equipment is expensive, the run-time is long, and it cannot be used at the point of care. In addition, during the ultracentrifugation, the exosomes might get damaged. The main advantage of microfluidic separation is the possibility of integration on the chip of downstream analysis to detect all the components of exosomes directly. Microfluidic immunoaffinity methods have resulted in highly purified exosomes but work only with cell-free samples, and therefore the cost might be high, primarily because of the cost of antibodies. Microfluidic devices are highly portable and easily integrated with micromixers and other parts. The success of the isolation depends on the quality of pre enriched exosomes.

We believe, as stated above, that the conventional methods will not be replaced by microfluidic devices, or maybe only for specific applications like the point-of-care where portability is necessary. It has to be stressed that lab-on-a-chip devices will evolve, and new functionalities will improve the quality and yield of microfluidic isolation. Magnetic immunocapture is an especially valuable method. Thus, some of them have already been commercialized as exosome isolation kits. Most important for the future of microfluidic research is the standardization and validation of methods that will allow accurate comparison between different laboratories.

In conclusion, researchers interested in working in this field should be aware of the existing challenges and be ready to use their skills and creativity.

## 5. Conclusions

In this review paper, exosomes and their importance as potential biomarkers for cancer diagnosis have been briefly discussed. Exosomes are incredibly complex biological species, carrying information from the cancer cells that release them as very early signals. Their isolation and subsequent detection methods are complicated and not satisfactory for the time being. However, they are evolving and enriched by new discoveries in nanoscience, microfluidics, plasmonics, etc. Significantly, new microfluidic techniques such as 3D and 4D printing will simplify the microfabrication techniques used to produce devices. At the same time, the exosome isolation and detection sensitivity could be enhanced by integrating several new designs of nanostructures with microstructures. The crucial importance of exosomes in cancer research is a powerful impetus for fast advances in isolation and detection techniques.

## Figures and Tables

**Figure 1 micromachines-13-00730-f001:**
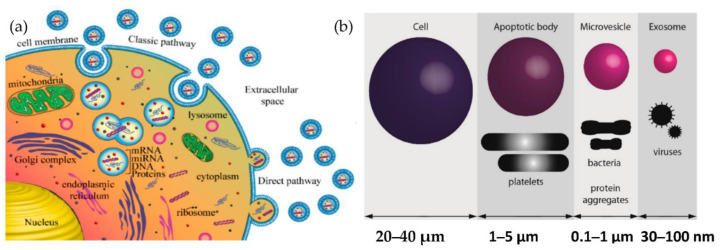
Biogenesis of extracellular vesicles (**a**) Exosome biogenesis (**b**) Comparison of the size of various types of extracellular vesicle with that of a cell.

**Figure 2 micromachines-13-00730-f002:**
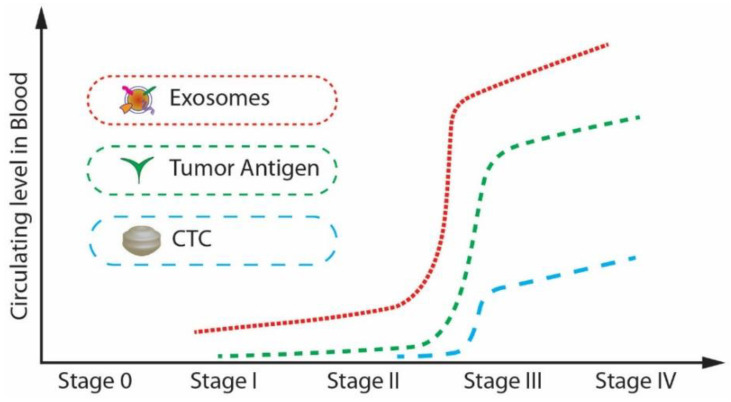
The circulating levels of tumour antigens, CTCs, and exosomes in blood during cancer progression. (Modified from He M., Zeng Y. [32]. Copyright 2016, J Lab Autom).

**Figure 3 micromachines-13-00730-f003:**
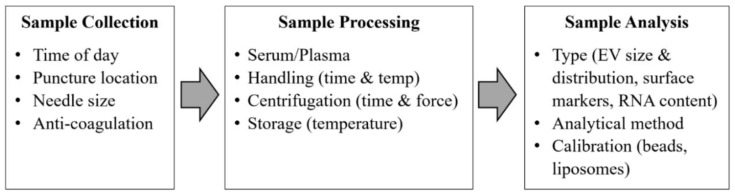
Critical standardization issues for exosomes analysis from blood samples. (Modified with permission from van der Meel R. et al. [37]. Copyright 2014, John Wiley and Sons).

**Figure 6 micromachines-13-00730-f006:**
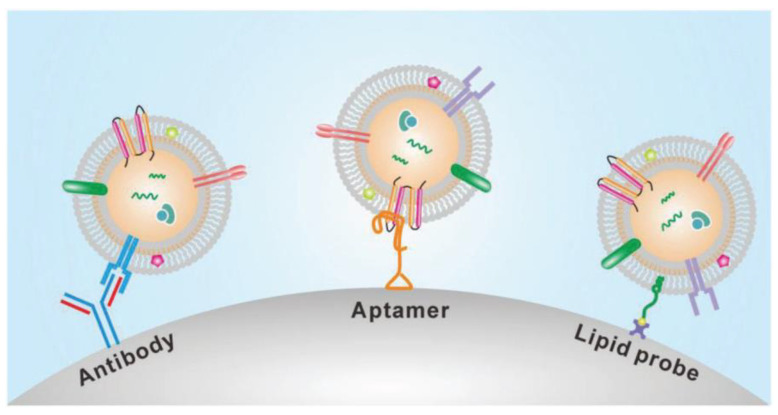
Immunoaffinity-based isolation of exosomes. (Reproduced with permission from Wang W. et al. [39]. Copyright 2018, John Wiley and Sons Inc.).

**Figure 7 micromachines-13-00730-f007:**
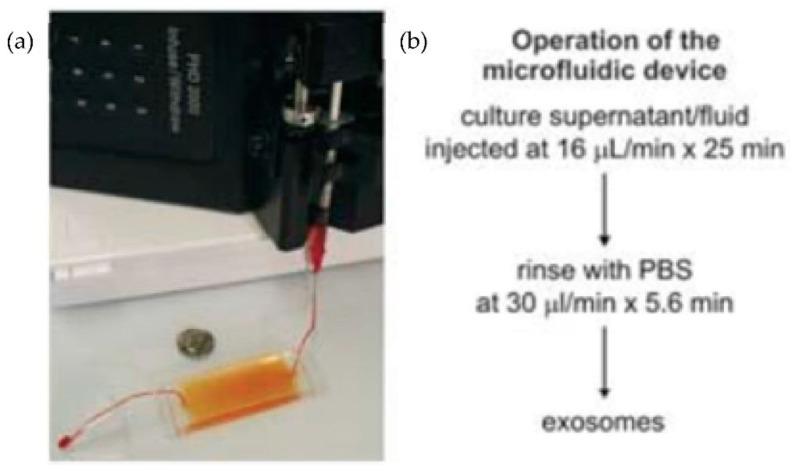
Experimental setup of microfluidic devices. (**a**) Photo of the device with a syringe pump. (**b**) Procedure for the isolation of microvesicles from the microfluidic device. (Reproduced from Chen et al. [31]. Copyright 2010, Lab Chip.).

**Figure 8 micromachines-13-00730-f008:**
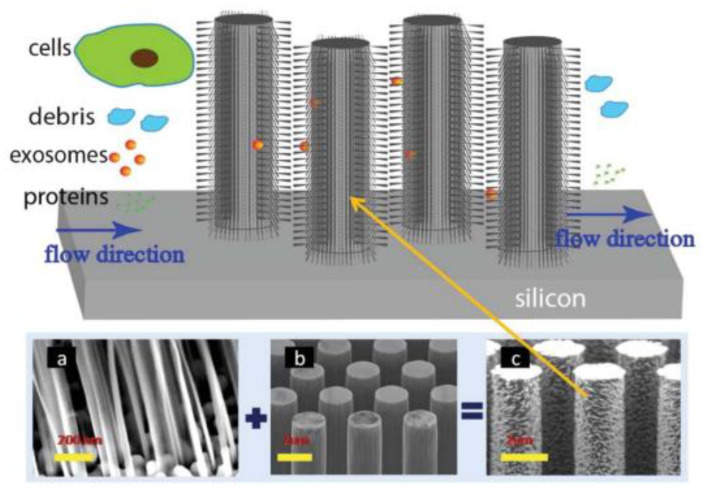
Schematic of the ciliated micropillar array. (**a**) representative porous silicon nanowire forest. (**b**) micropillars. (**c**) representative ciliated micropillars. (Reproduced from Wang Z. et al. [72]. Copyright 2013, Lab Chip).

**Figure 9 micromachines-13-00730-f009:**
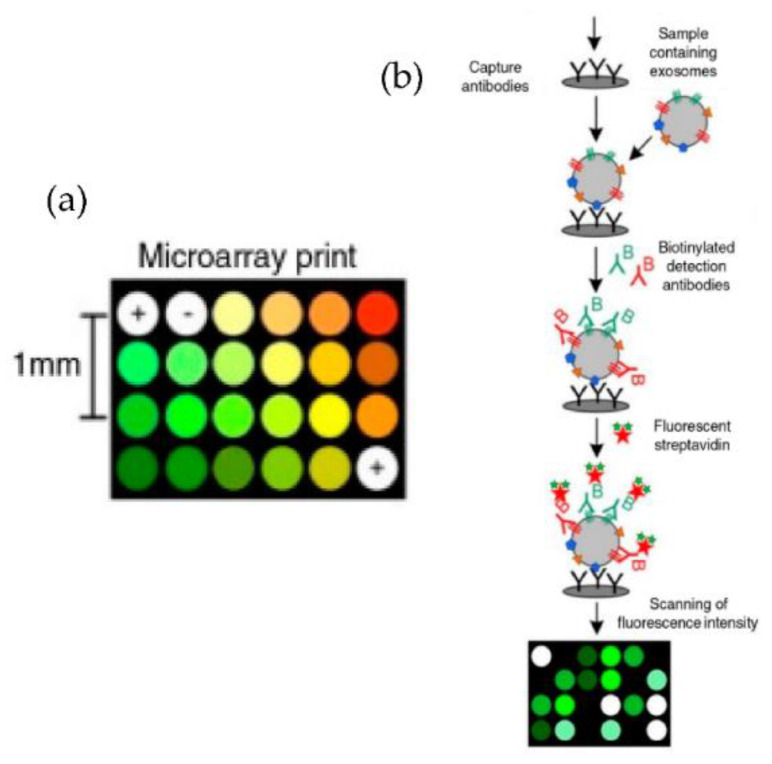
Extracellular vesicle detection using a customized protein microarray “EV array.” (**a**) A microarray printed with spots of 21 different antibodies. (**b**) Workflow for capturing exosomes with biotinylated antibodies followed by fluorescence-labelled streptavidin. (Modified from Jorgensen et al. [73]. Copyright 2013, J Extracell Vesicles).

**Figure 10 micromachines-13-00730-f010:**
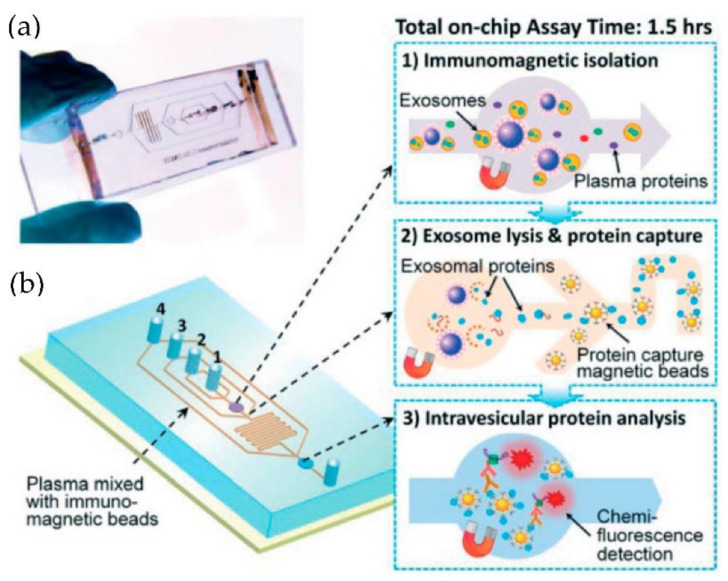
Integrated microfluidic exosome analysis directly from human plasma. (**a**) A photo of the prototype chip with the cascading microchannel for multi-stage exosome capture and analysis. (**b**) Schematic of the chip with the workflow. The numbers 1–4 indicate the inlet for exosome capture beads, washing/lysis buffer, protein capture beads, and ELISA reagents. (Reproduced from He M. et al. [75]. Copyright 2014, Lab Chip).

**Figure 11 micromachines-13-00730-f011:**
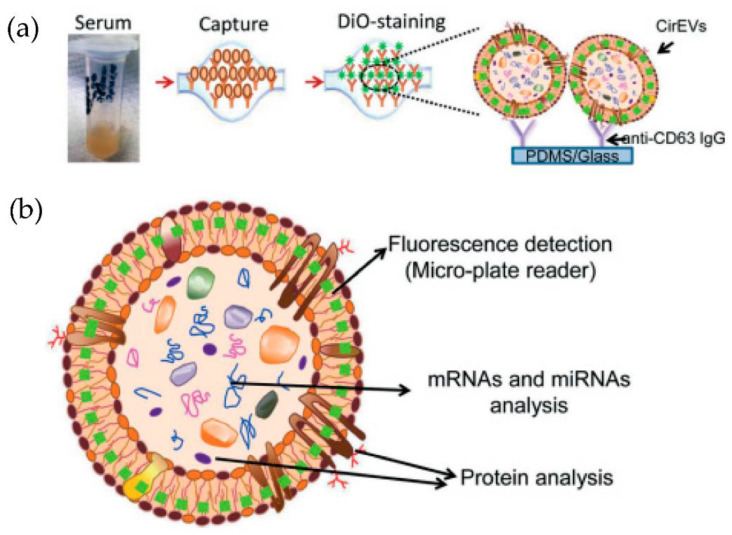
Experimental strategy for exosome immobilization and characterization using ExoChip (**a**) Schematic of exosome capture and analysis used in ExoChip. (**b**) Schematic for CirEVs fluorescence detection in micro-plate readers. (Reproduced from Kanwar S.S. et al. [76]. Copyright 2014, Lab Chip).

**Figure 12 micromachines-13-00730-f012:**
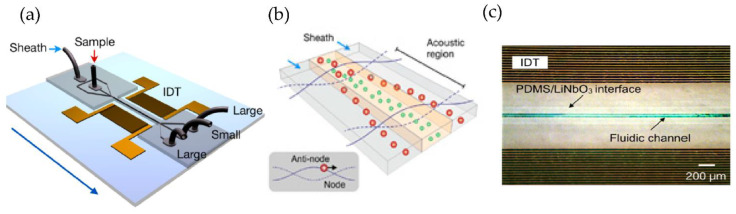
Acoustic nanofilter for label-separation of microvesicles (MVs). (**a**) Device schematic. (**b**) Filter operation indicates MVs transported to the acoustic pressure region (inset). (**c**) SEM image of the prototype device. (Adapted with permission from Lee K. et al. [77]. Copyright 2015, American Chemical Society).

**Figure 13 micromachines-13-00730-f013:**
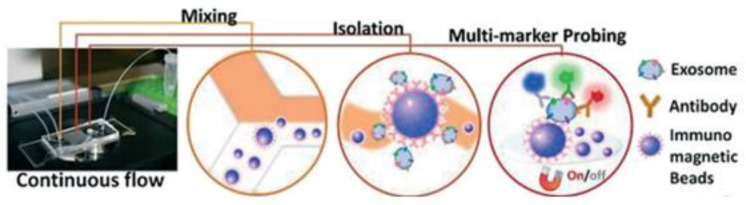
The setup and workflow of the ExoSearch chip for continuous mixing, isolation, and in situ, multiplexed detection of circulating exosomes. (Reproduced from Zhao Z. et al. [78]. Copyright 2016, Lab Chip).

**Figure 14 micromachines-13-00730-f014:**
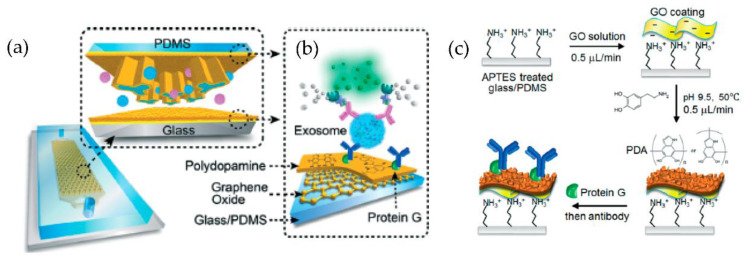
Nano-interfaced microfluidic exosome platform (nano-IMEX). (**a**) Schematic of a single-channel PDMS/glass device, with the exploded-view highlighting the coated PDMS chip containing an array of Y-shaped microposts. (**b**) Schematic showing the surface of the channel and microposts coated with graphene oxide (GO) and polydopamine (PDA). (**c**) The protocol involved in the surface functionalization of the microfluidic chips. (Reproduced from Zhang P. et al. [79]. Copyright 2016, Lab Chip).

**Figure 15 micromachines-13-00730-f015:**
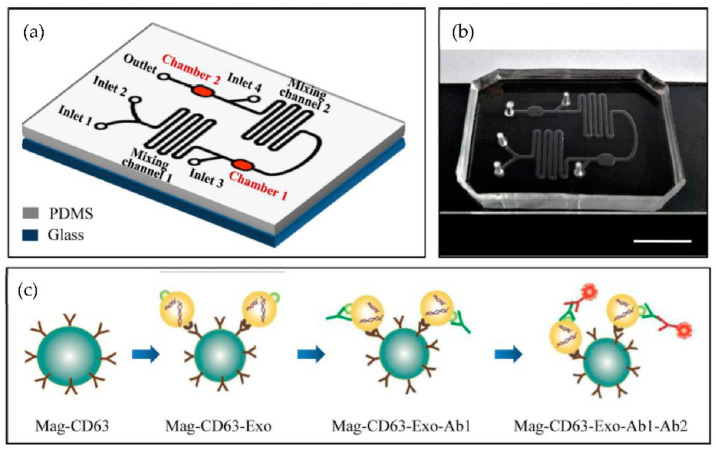
Chip design and principle for exosome capture and detection. (**a**) The microfluidic chip schematic representation. (**b**) Photo of the chip. The scale bar represents 1 cm. (**c**) Workflow for the immunomagnetic capture and detection of exosomes. (Reproduced from Fang S. et al. [80]. Copyright 2017, PLoS ONE).

**Figure 16 micromachines-13-00730-f016:**
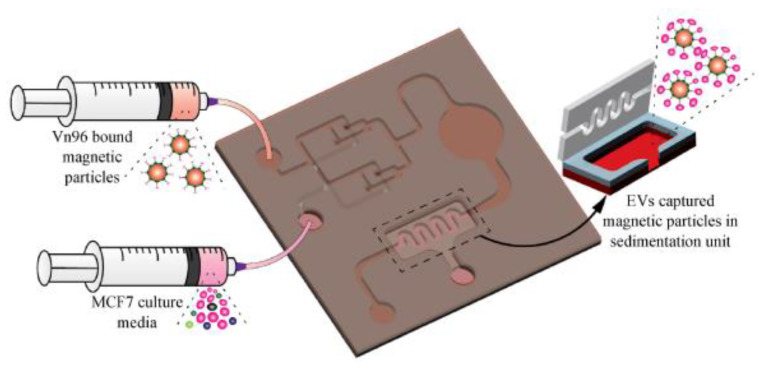
Isolation of EVs from the MCF7 CCM A) Schematic of the EV-isolation from the CCM in the microfluidic device. (Reproduced with permission from Bathini S. et al. [81]. Copyright 2021, Elsevier Science).

**Figure 17 micromachines-13-00730-f017:**
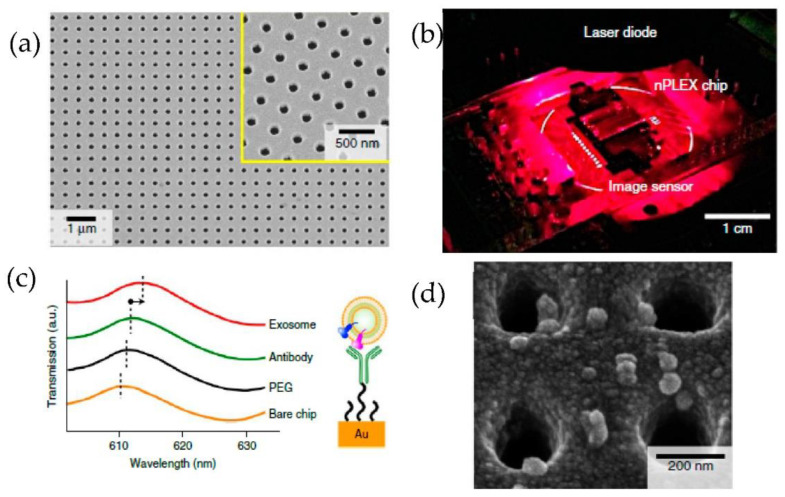
Label-free detection of exosomes with the nPLEX sensor. (**a**) A scanning electron image of the periodic nanoholes in the nPLEX sensor. (**b**) A prototype of the nPLEX imaging system. (**c**) Schematic to represent the changes in transmission spectra to show the exosome detection with the nPLEX device. (**d**) Scanning electron microscopy image showing the exosome capture near the nanohole array. (Reproduced from Im H. et al. [82]. Copyright 2014, Nat Biotechnol.).

**Figure 18 micromachines-13-00730-f018:**
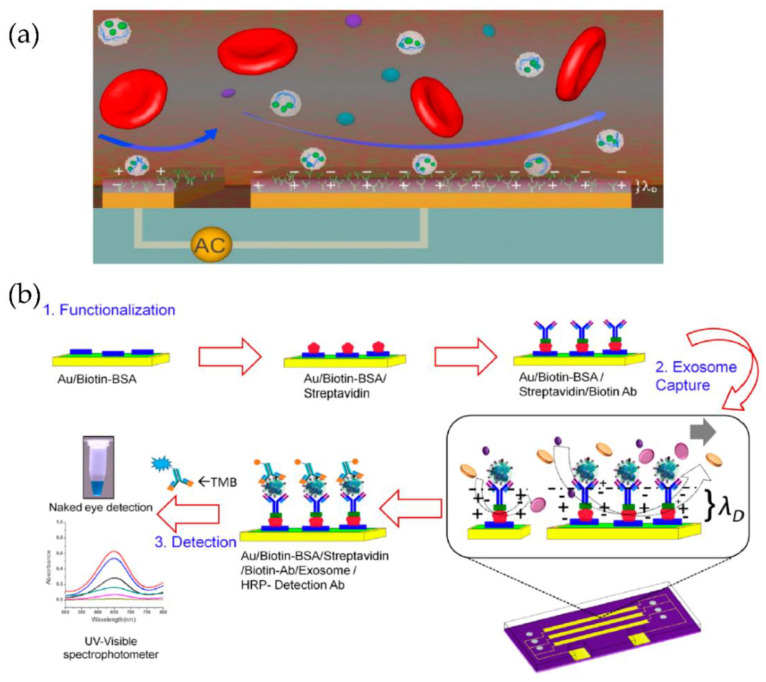
Tuneable alternating current electro hydrodynamic nanoshearing device for the detection of exosomes. (**a**) Schematic representation of ac-EHD induced device (appears as white spherical particles). (**b**) Schematic representation of the functionalization protocol. (Reproduced with permission from Vaidyanathan R. et al. [83]. Copyright 2014, American Chemical Society).

**Figure 19 micromachines-13-00730-f019:**
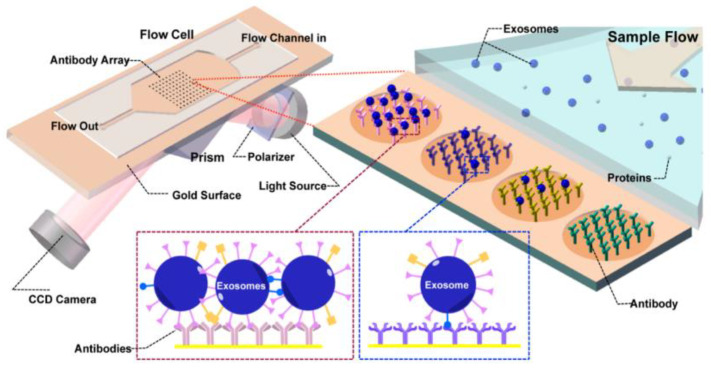
Label-free SPRi method for the detection and characterization of tumour-derived exosomes. Schematic view of SPRi combined with antibody microarray and measurement setup. (Reproduced from Zhu L. et al. [84]. Copyright 2014, Anal Chem.).

**Figure 20 micromachines-13-00730-f020:**
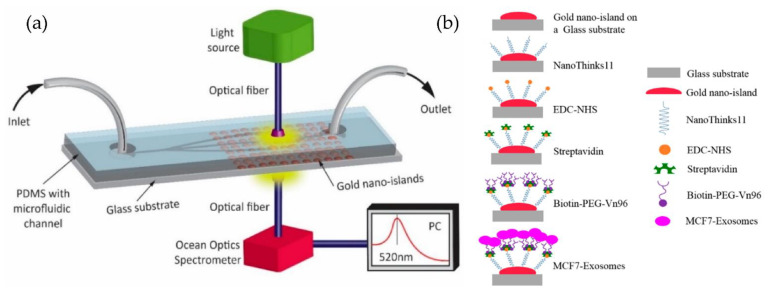
Label-free LSPR microfluidic device to detect MCF-7 exosomes. (**a**) Schematic of the microfluidic device with nano-island structures. (**b**) The biosensing protocol used to detect MCF-7 exosomes (Reproduced from Bathini S. et al. [86]. Copyright 2020, The European J Extracellular Vesicles).

**Figure 21 micromachines-13-00730-f021:**
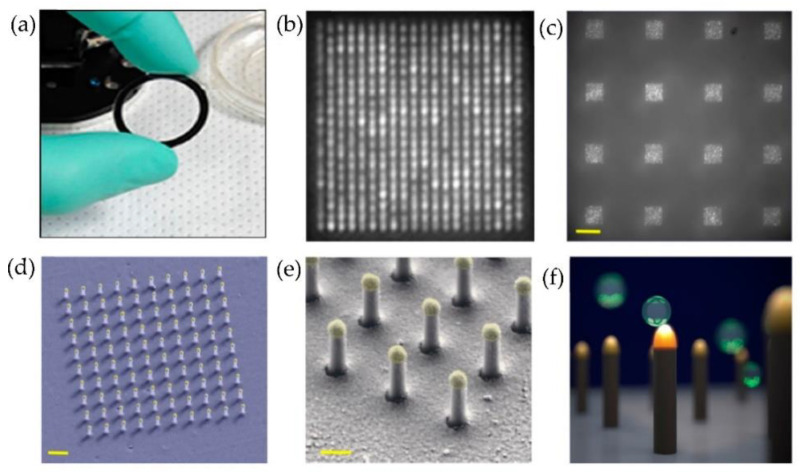
Nanoplasmonic pillars engineered for single exosome detection (**a**) Picture of the 25.4 mm diameter LSPRi sensor chip. (**b**) Scanning electron microscope image of the device for a 20 × 20 array, with a pitch size of 600 nm scale bar: 1 μm. (**c**) Image of sixteen arrays, each consisting of 400 plasmonic nanopillars in a 20 × 20 square lattice and 500 nm pitch, scale bar: 10 μm. (**d**) False coloured SEM image of a 10 × 10 nanopillar array, scale bar: 1 μm. (**e**) High-magnification false-coloured SEM image showing a detailed view of individual nanopillars, scale bar: 200 nm. (**f**) Picture illustrating the size matching of individual nanopillars diameter (d = 90 nm) to that of exosomes (~50 nm < d < 200 nm). (Reproduced from Raghu D. et al. [87]. Copyright 2018, PLoS ONE).

**Figure 22 micromachines-13-00730-f022:**
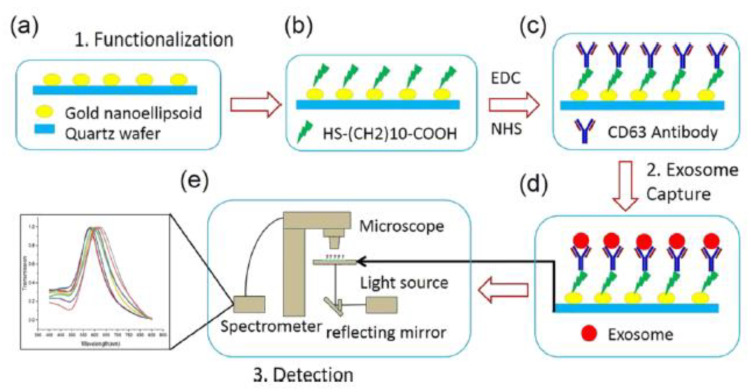
Schematic of the LSPR-based biosensor for the detection of the exosomes. (**a**) Fabricated gold nano-ellipsoid arrays on the quartz substrate. (**b**) Functionalization of the gold nano-ellipsoids with the Au–S bond. (**c**) Functionalization of anti-CD63 antibody (**d**) Exosomes injected into the microchannel and captured on the sensor substrate. (**e**) LSPR detecting platform (Reproduced with permission from Xiaoqing Lv et al. [88]. Copyright 2019, American Chemical Society).

**Table 1 micromachines-13-00730-t001:** Immunoaffinity approach integrated with microfluidics technique to capture and detect exosomes.

Techniques/Approaches	Markers Used for Detection	Sample Used and Its Volume	Detection Sensitivity (LOD)	Yield	Throughput of Isolation [µL/min]	Advantages	Disadvantages	Year of Work Published
Anti-CD63 functionalized surface with herringbone groves [31]	CD63	Serum of 100–400 µL	NA	42–94%	13.1	High specificity, Isolation time (~1 h)	Specific only for CD63	2010
An array of porous silicon nanowire-on-micropillars [72]	Liposomes (83, 120 nm)	Liposomes of 30 µL	NA	45–60%	10	Trapping is relatively fast (~10 min), high purity recovery of liposomes	Recovery time (~1 day), not validated with clinical samples, and no analysis of cargo protein	2013
Microarray spots (non-contact printing)—EV array [73]	CD9, CD63, CD81	Plasma of 1–10 µL	2.5 × 10^4^ exosomes per sensing spot	NA	NA	Multiplexed—24 analytes per array, highly sensitive and high-throughput	Isolation time (~3 days), the study carried out only on healthy donors	2013
Microarray spots (contact printing)—EV array [74]	60 markers simultaneously	Plasma of 1–10 µL	NA	NA	NA	Multiplexed - >60 analytes per array, higher sensitivity due to the contact printing	Isolation time (~3 days), the study carried out only on healthy donors	2015
Online mixing in a serpentine channel with immunomagnetic beads [75]	EpCAM,α-IGF-1R, CA125, CD9, CD81, and CD63	Plasma of 30 µL	0.28–0.38 pg/mL	42–97.3%	2	High specificity, isolation time (~1.5 h)	Specific for CA 125, EpCAM, and CD24	2014
An array of surface-functionalized circular microchambers ExoChip [76]	CD63 and extract total RNA	400 μL serum	0.5 pM	15–18 μg of total proteins	4	Easy scale-up, on-chip quantification	low capture capacity, no multiplexity	2014
Acoustic nanofilter chip [77]	Exosome markers: CD63,flotillin-1, HSP90,HSP70, microvesicles marker:β1-integrin	10 μL cell culture media and packed RBC	NA	80–90%	~0.24	90% separation yield, in situ control of size	Specific only for the microvesicles	2015
Multiplexed continuous mixing in a serpentine channel with immunomagnetic beads (ExoSearch) [78]	CA 125, EpCAM, and CD24	Plasma of 10 µL–10 mL	750 exosomes/μL	90%	0.8	Isolation time ~40 min	Specific for CA 125, EpCAM, and CD24	2016
Nano-IMEX microfluidic chip with Y-shaped microposts coated with (GO/PDA) [79]	CD9, CD63, CD81, EpCAM	Plasma of 2 µL	~50 exosomes/μL	NA	0.05	Enhanced efficiency, scalability	NA	2016
Microfluidic device integrated with immunomagneticocapture [80]	EpCAM, HER2	~1000 µL Cell culture medium and Patient plasma	NA	NA	2	Higher purity and intact yield	NA	2017
Microfluidic chip integrated with a 3D mixer and streptavidin coated magnetic particles [81]	HSP	0.2 mL of MCF 7 CCM EVs	NA	90%	NA	High yield, faster isolation time, isolation time ~20 min	specific only for HSP	2021

**Table 2 micromachines-13-00730-t002:** Immunoaffinity approach integrated with microfluidics technique and nanoplasmonic detections to capture and detect exosomes.

Techniques/Approaches	Markers Detected	Sample Used and Its Volume	Detection Sensitivity (LOD)	Yield	Throughput of Isolation [µL/min]	Advantages	Disadvantages	Year of Publication
Periodic Au nanohole arrays (nPLEX) chip [82]	CD45, CD63, CA125, CA19–9, D2–40, EpCAM, EGFR, HER2, CLDN3, and MUC18	Ascites of 150 µL	~3000 exosomes	NA	8.3	Isolation time (~30 min)	NA	2014
Microfluidic device with AC-EHD-induced [83]	HER2, CD9, PSA	Serum of 500 µL	~2760 exosomes/μL	NA	4.2	Multiplexed sensing, 3-fold enrichment in detection sensitivity compared to a normal hydrodynamic flow	NA	2014
Printed antibody microarray on an Au coated surface (SPRi) [84]	CD9, CD41, CD63, CD82, EpCAM, and E-cadherin	Cell culture supernatant (CCS) exosomes	~4.87 × 10^7^ exosomes/cm^2^	NA	NA	Real-time, label-free, and quantitative method	No multiplexity	2014
Au nano-island microfluidic device using LSPR [85,86]	HSP	100 µL of MCF7 cell culture media (CCM) exosomes	NA	NA	NA	Label-free technique	Specific for HSP	2018
Au nanoplasmonic array for LSPR based digitalized detection (LSPRi) [87]	CD 63	MCF7 secreted exosomes (1× 10^5^ exosomes/mL)	3 fold	NA	NA	Multiplexed measurements, one exosome can be detected and individually imaged in real-time	NA	2018
Nano-ellipsoid arrays integrated with a microfluidic chip using LSPR [88]	CD63	Lyophilized exosomes	1 ng/mL	NA	NA	Low-cost, time-saving, and applicable to large areas	NA	2019

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
