# Peer review of "Microfluidic Platforms for the Isolation and Detection of Exosomes: A Brief Review"

_micromachines, 2022, doi:10.3390/mi13050730_

Round 1
Reviewer 1 Report
Raju et al. have revised the literature related to the isolation of exosomes with microfluidic methods. Several papers have been taken into account, however many more papers have been published on this topic, in particular in the last few years. Punctual comments are listed below:
- The title claims for exosomes in cancer research, but only few of the reported studies isolate exosomes related somehow to cancer. The title seems not fully appropriated.
- The section 2 of the general part is aimed at describing functions and clinical applications of exosomes. However, the few lines of this section do not fulfil the ambitious promise of this part; therefore, it needs to be expanded or to be included in the introduction (after the paragraph describing the biogenesis of EVs).
- The section 3 describes the concept of liquid biopsy, but is limited almost to CDC and ctDNA. If it is true that these two components are the most studied, many other biomarkers are included in the term “liquid biopsy” and therefore they should at least be mentioned. This section is in general slightly confused and seems to break the logic of the review, which is centered to exosomes.
- The main part lists works that use microfluidic methods to isolate exosomes, but a critical discussion is missing. Moreover, recent papers on this topic should be included (or the selection of papers already present in the review should be better justified).
Reviewer 2 Report
- Raju et al completely explain what exosomes are and how they are released in two ways (classic pathway and direct pathway). Exosomes suggest that they may be helpful as vaccines for infectious diseases. Tumor biopsy could provide critical information on cancer but the limitation of any biopsy technique is that a single tumor may not capture all the mutations present. (Potential to help throughout all stages of cancer in general). The article is clear and organized. It is very informative and introductory to researchers who are interested in this field. The work is impressive and data is good enough to be published. Here are my suggestions:
- Authors could have one or two figures of their own, while the articles have 5 modified figures + 17 figures from others’ previous work.
- The comparison shown in Table 1 and 2 can be more informative. ‘Year of work published’ may not be necessary. However, specificity, or processing time, or samples used are more informative, and should be discussed/compared in the tables.
- Section 5.2 Lab-on a Chip (LOC)/microfluidics for isolation of exosomes
The first two paragraphs, which mainly discuss the LOC’s advantage, are too long, while this session is intended to discuss the exosome isolations. Authors should condense these two paragraphs, and focus on the isolation discussion.
The authors are suggested to explain how the isolation technologies can be grouped into these two sub-sessions. Please also revise the two sub-sessions and to organize the technologies discussed in the sub-session (instead of just describing the technology details) so the readers can really see how the technologies are grouped into these two. In addition, the titles for these two sub-sessions should be paralleled or consistent. (e.g. approach vs. methods; Immunoaffinity vs Immune-affinity)
5.2.1: Immunoaffinity approach to capture and detect exosomes
5.2.2: Immuno-affinity methods with nanoplasmonic detection to capture and detect exosomes
- Authors are suggested to discuss a few difference or cons/pros of these technologies after Table 1 and 2.
- Authors are suggested to have a few outlooks or guidelines in the manuscript before the 9. Summary session, so the readers can begin doing researches in this field. For instance, what could be promising directions and challenges (in sample preparations, or technology limitations aspect, or throughput, or yield etc.), in using lab-chip devices for exosome detection?
- There are a few questions for some of the readers are interested after reading this article:
- Researchers design lab-chip devices to replace traditional methods to measure physical properties of exosomes. While the traditional methods can still hold higher detection accuracy (or yield) than lab-chip devices, what is the forward-looking and possibility of applying lab-chip onto exosomes?
- Can the fabrication process to delicate and complex chips be applicable to mass production?
- The authors are also encouraged to include a few related references from MDPI’s “micromachines”
Round 2
Reviewer 1 Report
After revision, the authors improved their manuscript adequately and now the paper can be published in the present form.
Reviewer 2 Report
The authors have made all changes and greatly improved the manuscript. I recommend this article should be accepted.